# Impact of On-Farm Interventions against CTX-Resistant *Escherichia coli* on the Contamination of Carcasses before and during an Experimental Slaughter

**DOI:** 10.3390/antibiotics10030228

**Published:** 2021-02-24

**Authors:** Michaela Projahn, Jana Sachsenroeder, Guido Correia-Carreira, Evelyne Becker, Annett Martin, Christian Thomas, Carolin Hobe, Felix Reich, Caroline Robé, Uwe Roesler, Annemarie Kaesbohrer, Niels Bandick

**Affiliations:** 1German Federal Institute for Risk Assessment, Diedersdorfer Weg 1, 12277 Berlin, Germany; jana.sachsenroeder@bfr.bund.de (J.S.); guido.correia-carreira@bfr.bund.de (G.C.-C.); evelyne.becker@bfr.bund.de (E.B.); annett.martin@bfr.bund.de (A.M.); christian.thomas@bfr.bund.de (C.T.); carolin.hobe@bfr.bund.de (C.H.); felix.reich@bfr.bund.de (F.R.); annemarie.kaesbohrer@bfr.bund.de (A.K.); niels.bandick@bfr.bund.de (N.B.); 2Institute for Animal Hygiene and Environmental Health, Freie Universität Berlin, Robert von Ostertag-Straße 7-13, 14163 Berlin, Germany; caroline.robe@fu-berlin.de (C.R.); Uwe.Roesler@fu-berlin.de (U.R.)

**Keywords:** broiler, chicken, control measures, ESBL, pAmpC, slaughter, *Enterobacteriaceae*, farming conditions, stocking density, competitive exclusion, CTX-resistance

## Abstract

Cefotaxime (CTX)-resistant *Enterobacteriaceae* are still an ongoing challenge in human and veterinary health. High prevalence of these resistant bacteria is detected in broiler chickens and the prevention of their dissemination along the production pyramid is of major concern. The impact of certain on-farm interventions on the external bacterial contamination of broiler chickens, as well as their influence on single processing steps and (cross-) contamination, have not yet been evaluated. Therefore, we investigated breast skin swab samples of broiler chickens before and during slaughter at an experimental slaughter facility. Broiler chickens were previously challenged with CTX-resistant *Escherichia coli* strains in a seeder-bird model and subjected to none (control group (CG)) or four different on-farm interventions: drinking water supplementation based on organic acids (DW), slow growing breed Rowan × Ranger (RR), reduced stocking density (25 kg/sqm) and competitive exclusion with Enterobacteriales strain IHIT36098(CE). Chickens of RR, 25 kg/sqm, and CE showed significant reductions of the external contamination compared to CG. The evaluation of a visual scoring system indicated that wet and dirty broiler chickens are more likely a vehicle for the dissemination of CTX-resistant and total *Enterobacteriaceae* into the slaughterhouses and contribute to higher rates of (cross-) contamination during processing.

## 1. Introduction

The occurrence of multidrug resistant commensal *Escherichia (E.) coli* in broiler chickens is an ongoing threat to public health. Especially extended-spectrum beta-lactamase (ESBL) and plasmid-mediated cephalosporinase (pAmpC) producing *E. coli* are of major concern as they show resistance to the 3rd generation cephalosporin Cefotaxime (CTX). Although these bacteria might not affect the health of the broiler chickens, they are spread along the broiler production chain [1,2] and can contribute to the dissemination of antibiotic resistance determinants into the environment [3].

To decrease the prevalence of ESBL/pAmpC-producing *E. coli* in broiler chickens and chicken meat, intervention measures at the different stages of the production chain are necessary. On-farm interventions may comprise measures concerning the animal-related environment as well as the chicken itself. Studies showed that inappropriate cleaning and disinfection of stables can contribute to the colonization of chickens with resistant bacteria from previous flocks [4,5]. Treatment of chickens with feed additives [6] or the application of competitive exclusion (CE), could reduce the colonization and excretion of ESBL/pAmpC-producing *E. coli* [7,8]. However, the impact of a certain intervention measure on the level of contamination of the chicken before slaughter, during slaughter and the carcass after slaughter, have not been investigated.

Currently there are no studies published concerning specific interventions against ESBL/pAmpC-producing *E. coli* during processing in the slaughterhouse [9]. Furthermore, the magnitude of external contamination of broiler chickens (feathers, skin and feet) before processing is suspected to contribute to the contamination of carcasses in the slaughterhouses [10,11]. In particular, scalding and defeathering were shown to have an impact on cross and recontamination of broiler carcasses [12,13,14,15,16]. It is, therefore, reasonable to assume that the housing conditions of broiler chickens and respective measures against the colonization with ESBL/pAmpC-producing *E. coli* can have an impact on the external contamination of the bird with feces and, subsequently, on the contamination level of the chicken meat with these resistant bacteria.

Studies concerning the occurrence of resistant and/or pathogenic bacteria in broiler chickens or chicken meat usually focus on the farm level or the slaughterhouse level only. In this study, both stages of broiler production were combined to investigate the impact of on-farm interventions against CTX-resistant *E. coli* on the level of (cross-) contamination of carcasses after transport, before, during and after slaughter. Furthermore, we examined the level of external bacterial contamination of the broiler carcasses during slaughter. A scoring system was implemented for the visible external contamination of the breast skin area as a parameter for the outer contamination of the birds.

## 2. Results

### 2.1. Experimental Slaughter

Broiler chickens experimentally infected with CTX-resistant *E. coli* were fattened for 34 to 47 days depending on the breed (Table 1). One control group (CG) and four intervention groups (drinking water supplementation based on a mixture of organic acids (DW), slow growing breed Rowan × Ranger (RR), reduced stocking density of 25 kg/sqm (instead of 39 kg/sqm) and competitive exclusion with a single Enterobacteriales strain IHIT36098 (CE)) were investigated in this study. Transportation of the broiler chicken from the experimental animal facility to the slaughter facility lasted about 30 minutes. Maximum air temperature on the respective day of slaughter varied according to season and month (Table 1). Due to the use of a small scalding kettle, the mean water temperature ranged from 65 to 73 °C between the experimental groups. The mean weight of the chickens for slaughter was chosen to be 2 kg according to the experimental trial conditions. After evisceration the mean weight of the carcasses of the groups differed by 90 g, except for the 25 kg/sqm group where the mean weight was only 1889 g. The mean weight of the internal organ bundles varied between 239 g and 266 g.

### 2.2. CTX-Resistant Enterobacteriaceae

Swab samples from the breast skin of the broiler chickens were investigated quantitatively concerning CTX-resistant *Enterobacteriaceae* at four different stages before and during slaughter (after the arrival at slaughter (ST1), after scalding (ST2), after defeathering (ST3) and after evisceration (ST4)). Corresponding descriptive statistics are displayed in Table 2. The numbers of positive swab samples varied between the different stages but also between the different treatment groups. Groups CG and DW had the highest number of positive swabs at all four tested stations (32–40), whereas groups 25 kg/sqm and CE had the lowest number of positive samples (3–23). In the RR group, a reduction of positive swabs (from 31 in ST1 to 13 in ST4) along the slaughter process was observed. Overall, the mean concentration of CTX-resistant *Enterobacteriaceae* ranged between 0 and 4.72 log10 CFU/20 cm² breast skin. The counts of CTX-resistant *Enterobacteriaceae* were generally higher in the CG and DW groups than in the RR, 25 kg/sqm or CE groups. 

Quantitative results of the breast skin swabs concerning CTX-resistant *Enterobacteriaceae* showed significant reductions at the 0.1% level between the control group CG and the three treatment groups RR, 25 kg/sqm and CE, at all stations using a nonparametric multiple contrast test. In contrast, no significant difference at the 5 % level was determined between the control group CG and the DW treatment group at each of the stations (Figure 1). However, when performing a less conservative Conover-Iman-test with Holm correction for multiple testing, we found significantly different CFU/20 cm² in the DW group at the 5 % level at all stations (results are provided in the Appendix A to this paper). 

### 2.3. Total Enterobacteriaceae

In addition to CTX-resistant *Enterobacteriaceae* the concentrations of *Enterobacteriaceae* from the breast skin swab samples were investigated. Corresponding descriptive statistics are shown in Table 3. Numbers of quantitative positive swabs varied between 25 and 40 for all stations. Only on ST2 for the groups RR and 25 kg/sqm were low numbers of positive samples (three and nine, respectively) determined. 

Significance levels for the four treatment groups differed at all four stations (Figure 2). Only for treatment group 25 kg/sqm was there a significant reduction compared to CG determined for all of the investigated stations (ST1-4).

### 2.4. Scoring of Visible Contamination of Broiler Chickens

The Federal Ministry of Food and Agriculture recently revised the general administrative regulation for the section of food hygiene highlighting the impact of visible contamination with feces and litter of farm animals when entering the slaughterhouse (www.bundesanzeiger.de, AAV LmH BAnz AT 23.07.2019 B2) on contamination during the slaughter process. However, there is no scoring system available or defined for this matter.

In our study, immediately after arrival at the experimental slaughter facility, broiler chickens were subjected to a visual inspection with photographic documentation of the breast skin concerning fecal contamination. This was carried out for all chickens in all groups except for DW, where only a few photos were taken. Examples of the contamination with feces and status of feathers are shown in Figure 3. The pictures taken underwent a scoring process based on the appearance of feathers and coat (range 0 to 3) as well as the visible dirt/fecal contamination (range 0 to 2) and the mean values were calculated for each group (Table 4). 

Looking at the mean scores for fecal contamination of the breast skin, the feathers and the sum scores show variation between the groups, but also between the broiler chickens within a respective group (Table 4). In the 25 kg/sqm group, chickens had better feather scores, i.e. not as wet and dirty as in the CE and CG groups. Broiler chickens of the slow growing breed RR generally showed a different appearance due to their long and brownish colored feathers, but there was also a high level of contamination visible (Figure 3). Along with this observation based on the mean scores shown in Table 4, nonparametric statistical tests were performed in order to quantify the statistical significance of differences between the experimental groups (see Appendix A for details). The tests showed, in general, that there were statistically significant differences between the CG and the other experimental groups for the mean feather score. However, for the fecal contamination score a significant difference was only found between the CG and the 25 kg/sqm group and the CE group. Differences between CG and RR were not found to be statistically significant. Interestingly, in the RR group the fecal contamination comprised mainly large, loosely attached contaminations (score 2b), whereas in groups CE and CG higher amounts of smaller sized fecal contaminations dominated (score 2a). Six persons performed the scoring independently. Mean values for each experimental group varied between the different scorers from approximately 0.5 to 1 points, but the rankings of the groups (clean to dirty) were identical (Table 5).

Scoring values of each broiler chicken were then related to the bacterial counts of CTX-resistant and total *Enterobacteriaceae* (Figure 4). Combined scoring values show a general association between the score and the measured bacterial counts (Figure 4). Correlations between the total score and the log10 values of CTX-resistant and total *Enterobacteriaceae* were calculated to be 0.68 and 0.62, respectively. High scores with values of four and five seemed to be mainly associated with at minimum 10^3^ CFU/cm² of CTX-resistant, and at minimum 10^4^ CFU/cm² total, *Enterobacteriaceae*. However, there were outliers with high scoring values and low bacterial counts and vice versa. There also seemed to be some group specific differences. For example, when comparing the CE and the RR groups there was a notable difference in scoring (2.90 vs. 3.86), whereas there was only a slight difference in the initial contamination with total *Enterobacteriaceae* (4.49 vs 4.38) upon the arrival at the experimental slaughter facility (ST1).

### 2.5. Impact of the Scalding Process on the Bacterial Reduction

Scalding temperatures were measured with a temperature data logger or a thermometer and were assigned to the respective broiler chicken being processed, where possible. In addition to preparing for plucking, the scalding process should also represent a microbiological hurdle. Measured temperatures varied between the groups but were all at minimum 10 °C higher than typically used in a processing plant, as we also wanted to evaluate recontamination via processing. The mean reduction of CTX-resistant *Enterobacteriaceae* ranged from 0.18 to 3.46 log10 CFU/20 cm² on breast skin, and the concentration of total *Enterobacteriaceae* was lowered by 0.32 to 3.43 log10 CFU/20 cm² on breast skin (Table 6). 

We did not find any correlation between the mean values of the initial bacterial load at ST1, the mean reduction values and the mean scalding temperatures. However, when looking at the single bird data we found two main factors influencing the reduction of the bacterial load (note that for some birds an increase in bacterial load was observed as indicated by negative reduction values). Firstly, there was an indication that the amount of reduced bacteria was dependent on the initial load on the breast skin as shown in Figure 5. Secondly, the reduction via the scalding process seemed also to be dependent on the temperature because, in the plot, a second cluster appeared which mainly comprised carcasses of the groups DW and CG, which were scalded at mean temperatures just below 70 °C. These correlations were determined for both CTX-resistant and total *Enterobacteriaceae* (Figure 5, Appendix A). 

Two linear models additionally confirmed observations. The results are shown in Table 7. The models indicated that temperature and initial bacterial load were important factors for predicting the log reduction due to scalding. Details on the model selection are provided in the Appendix A. We wish to emphasize here that for the data on all *Enterobacteriaceae* the models with a comparable number of independent variables led to mostly nonsignificant estimates, which is why the model based on the data on all *Enterobacteriaceae* contained fewer independent variables than the model based on data of resistant bacteria. 

### 2.6. Investigation of Defeathering on the Bacterial Recontamination

Scalding temperatures were set at high values to achieve a large reduction of bacterial concentrations on carcass surfaces. This approach was applied to improve determination of the extent of recontamination during defeathering. Mean changes on the bacterial load on the breast skin are shown in Table 8.

When plotting single bird values after scalding against the changes after defeathering, it was observed that for most birds of the highly contaminated groups CG and DW there was a further reduction of up to 3 log10 values (Figure 6). In contrast, more than half of the carcasses, especially from the RR and 25 kg/sqm groups, were recontaminated with up to 4 log10 units as indicated by a negative reduction.

## 3. Discussion

In this study, we investigated the impact of possible on-farm measures against ESBL/pAmpC-producing *E. coli* on the external contamination of broiler chickens (breast skin), as well as the amount of cross-contamination of carcasses during slaughter, in an experimental slaughter facility. Furthermore, we developed a scoring system for the external contamination of the breast skin area of the broiler chickens based on the appearance of the feathers/coat and the visible contamination with dirt and feces. 

### 3.1. Impact of Processing Steps on Bacterial Contamination 

We found the external contamination of the carcasses to be reduced by 1 to 2 log10 values during processing, which compares well with published data from commercial slaughterhouse facilities [17]. The reductions obtained after the scalding step (ST2) differed between the treatment groups. Possible factors for the varying reductions could be the scalding temperature as well as the initial contamination rate at ST1. In our experiments, we choose high scalding temperatures to maximally reduce the bacterial load on the carcasses for investigating the recontamination during the defeathering process. These temperatures exceeded those usually used for low-temperature scalding at commercial slaughterhouses, and we found the effectiveness of the scalding process to be dependent on the initial bacterial load as well as the temperature levels as calculated by the models. This was shown for both CTX-resistant and total *Enterobacteriaceae*. It was previously demonstrated that the inactivation of *E. coli* on breast skin is temperature dependent in the low-temperature scalding range [18]. However, to further assess the dependency of scalding temperature and initial load on the bacterial load after the process, investigation under practical conditions in slaughterhouses is necessary. 

The defeathering process led to a further reduction of about 1 log10 values in the highly contaminated groups (CG, DW) but resulted in a recontamination of up to 2.5 log10 values in the low contamination groups (RR, 25 kg/sqm, CE). Pacholewicz et al. found that effects of the defeathering process differ among slaughterhouses and may lead to either a reduction of or an increase in bacterial concentration [19]. Our results indicate the initial contamination of the carcasses as another potential factor for the effect of defeathering on bacterial load. This latter point has been suggested before [20,21]. Furthermore, it seems that recontamination is more easily detected when the initial overall contamination is low. In cases of high initial bacterial load, the subsequent recontamination no longer contributes significantly to an increase in the overall bacterial load on the carcass. It was previously shown that the defeathering process in commercial slaughterhouses can contribute to contamination of the carcasses with resistant bacteria [15,16]. 

In our study, the magnitude of recontamination of the carcasses during evisceration was limited. This might be due to the manual evisceration step and needs to be further evaluated in a large scale study.

### 3.2. Peculiarity of the DW Group 

Large numbers of resistant bacteria, although not significantly different at the 5 % level from the control group CG, were detected in the DW group after the arrival at the slaughter facility, i.e. at ST1 in our experimental slaughtering process. One factor that might account for this observation is that the drinking water supplement used was developed to increase the appetite of the animals. This stimulated increased food intake and subsequently increased gut activity and increased fecal excretion. As the litter in the broiler fattening barns was not changed during the fattening period, an increased fecal excretion led to increased fecal contamination of the litter, which in turn may have led to a higher external contamination of the broiler chickens sitting and sleeping on this litter. Along the processing stations, the pattern of statistically significant differences between the control group and the DW group showed some variation in the level of contamination with CTX-resistant and total *Enterobacteriaceae*. While for CTX-resistant bacteria no differences at the 5 % level between control group and DW group were found at any station, there were statistically significant differences at the 5 % level for total *Enterobacteriaceae* at stations 2 to 4.

### 3.3. Bacterial Contamination throughout the Experimental Groups 

The treatment groups RR, 25 kg/sqm and CE showed significantly lower contamination rates with CTX-resistant *Enterobacteriaceae* on the breast skin compared with the control group at the four investigated processing stages. Lower concentrations of bacteria after arrival (ST1) were determined in the 25 kg/sqm group whereas, after evisceration (ST4), the RR group was least contaminated. Variation in the amounts of CTX-resistant bacteria on the breast skin of the broiler chickens of these three treatment groups might be due to the different processing days and, therefore, slightly different processing conditions influenced by personnel, temperature during transport, waiting period and scalding temperature. The lower contamination rates of the RR broiler chickens are in line with the different appearance of these birds as indicated by the visual scoring. It might be that their more abundant feathers provided a certain protection to the breast skin from contamination. It is already known that the surface of the chicken skin has a great impact on contamination and that bacteria are easily attached to the skin and protected from removal by associated polymers [18,22]. The initial concentration of total *Enterobacteriaceae* at ST1 was slightly reduced for RR although not significantly, supporting the hypothesis that the increased density of feathers affords some protection against contamination. The lowest concentrations of CTX-resistant and total *Enterobacteriaceae* on the breast skin at ST1 were determined for the 25 kg/sqm group. A straightforward explanation for this would be that fewer chickens produce less feces and, therefore, the litter and, consequently, the chicken breast skin become less contaminated. The feathers of the chickens appeared less dirty and wet in the 25 kg/sqm group as compared to the other groups. In addition to the overall external contamination, one can assume that the reduced stocking density might have an impact on the colonization and the spread of the resistant bacteria between the chickens during the fattening period, as indicated previously [23]. 

Broiler chickens of the CE group also showed reduced external contamination rates with CTX-resistant and total *Enterobacteriaceae*. In this treatment, group chickens were inoculated with only a defined, single Enterobacteriales strain (IHIT36098) before inoculation with the resistant bacteria in the seeder-bird model. CE cultures were originally developed to prevent gut colonization with enteropathogenic bacteria e.g. *Salmonella* [24,25,26]. Recent studies showed that complex, nondefined CE cultures could reduce colonization and shedding of ESBL-*E. coli* dependent on the study and the resistant strains investigated [8,27]. However, in our study there were still detectable contaminations with CTX-resistant bacteria, which indicates that the CE culture did not provide full protection from colonization with all resistant bacteria. This might underscore findings from other studies where varying effects were found for the reduction of the colonization or shedding of CTX-resistant bacteria, depending on the type of resistant strains employed [28,29]. The effect of a defined CE culture composed of several apathogenic microorganisms of different bacterial families, its protective effects on the colonization of broilers with CTX-resistant bacteria, and subsequently on the contamination at slaughter, needs to be further evaluated.

In our study, we observed no differences in total *Enterobacteriaceae* contamination between the CE and CG groups after the evisceration step (ST4). This was also the case for the 25 kg/sqm group. However, broiler chickens of the CE group had the second best visual scoring value, indicating a less dirty appearance than the chickens in the CG and RR groups.

### 3.4. Evaluation of the Scoring System

We also evaluated the correlation of the general appearance of the broiler chickens with the bacterial contamination of the breast skin. For this, we implemented and evaluated a visual scoring system based on photographs of the birds taken directly after arrival at the experimental slaughter facility. The ranking of the final mean scores paralleled the mean bacterial contamination of the breast skin with both CTX-resistant and total *Enterobacteriaceae*. We did not photograph the chickens of the DW group, which had the highest bacterial loads. The comparison of the individual birds of the other four groups showed that the dirtier and wetter the chickens appeared, the more difficult it was to be to classify them correctly. However, as the overall aim should be to reduce the amount of contamination that enters the slaughterhouse via the chickens, the total score should be as low as possible. In Germany, the Federal Ministry of Food and Agriculture recently revised the general administrative regulations for the section of food hygiene, highlighting the impact of visible contamination of farm animals with feces and litter when entering the slaughterhouse (www.bundesanzeiger.de, AAV LmH BAnz AT 23.07.2019 B2). However, there is currently no official scoring system, such as for the scoring of foot dermatitis and lesions [30]. In our study, we tested a scoring system for the breast skin area of broiler chickens comprising characteristics of appearance of feathers as well as the contamination with dirt and feces. Independent scoring showed clear differences between the four tested groups. However, absolute values varied between the scorers. This indicates that scoring of the visible contamination of broiler chickens is possible and that low values are very likely associated with low contamination levels of CTX-resistant and total *Enterobacteriaceae*. However, the description for each category, and the respective scores, need to be further defined and evaluated to ensure consistency. Furthermore, scoring difficulties due to particular differences between breeds must be accounted for.

## 4. Materials and Methods 

This animal study was permitted by the Berlin State Office of Health and Social Affairs, Berlin, Germany (proposal number 0193/16).

### 4.1. Bacterial Strains

Two different CTX-resistant *E. coli* strains were used for the infection trials [23]. Strain 10716 harbors a *bla*_CTX-M-15_ and is assigned as MLST ST-410 whereas strain 10717 harbors the *bla* genes for a TEM-1 beta-lactamase and a CMY-2 plasmid-encoded cephalosporinase (pAmpC beta-lactamase) and is assigned as MLST ST-10.

### 4.2. Colonization Trials of Broiler Chickens

Hatching, raising and infection trials of broiler chickens using the seeder-bird model were performed by the Institute for Animal Hygiene and Environmental Health of the Freie Universität Berlin and conducted in the animal facilities of the Centre for Infection Medicine of the Department for Veterinary Medicine of the Freie Universität Berlin. Eggs from ESBL/pAmpC free parent Ross 308 flocks were disinfected twice using formaldehyde gas and WESSOCLEAN^®^ K 50 Gold Line (Wesso AG, Hersbruck, Germany) and were incubated in a small-scale hatcher (Heka Favorit Olymp 432, Rietberg, Germany) for 21 days. Ninety broiler chicks (18 seeder + 72 sentinels) were conventionally housed in the animal facility under identical conditions independent of the season and provided with conventional feed and water ad libitum, with a stocking density of 39 kg/sqm. Oral inoculation of the 18 seeder birds with 200 µL of the CTX-resistant strains (2 log10 CFU/mL each) was performed on day three after hatching. Broiler chickens were kept together the whole time and were fattened to an average weight of 2 kg/bird, which corresponded to a fattening period of 34 to 38 days for the Ross308 breed and 47 days for Rowan × Ranger. One control group (CG) and four intervention groups were investigated in this study: drinking water supplement (DW), slow growing breed Rowan × Ranger (RR), reduced stocking density (25 kg/sqm) and competitive exclusion (CE). These intervention measures were applied throughout the fattening period as previously described [23]. In brief, the DW group received a water a supplement based on a mixture of organic acids widely used in broiler production added to the drinking water three times during the rearing cycle as recommended by the manufacturer (fattening days 2–7, days 15–19, days 31–38). For the reduced stocking density, the pen area in the fattening facility was enlarged accordingly. In the CE group all broilers were orally inoculated with 8 log10 CFU/ml of a single, nonpathogenic Enterobacteriales strain (IHIT36098) on the day of the hatching, one day before the oral inoculation of the CTX-resistant challenge strains. The CE strain IHIT36098 was isolated and characterized by the Institute of Hygiene and Infectious Diseases of Animals, Justus Liebig University Giessen, Germany.

### 4.3. Broiler Processing

After the respective fattening time, 40 sentinel chicken were transported to the experimental slaughter facility of the German Federal Institute of Risk Assessment, Berlin, Germany, subjected to the experimental slaughter process and investigated concerning the level of external contamination with CTX-resistant *Enterobacteriaceae* before and during the processing (after evisceration). After arrival at the experimental slaughter facility, broilers were electrically stunned (manually handled stunning tong, moistened sponge electrodes, duration 8 s, 240 mA and 50 Hz) and bled by severing the carotid artery and jugular vein. During bleeding, the carcasses were placed in a funnel to prevent flapping of the wings. Broilers were subjected to high temperature scalding at a minimum temperature of 65 °C. This temperature was higher than that usually applied in commercial broiler processing but was chosen to significantly lower bacterial contamination of skin and feathers. This was necessary to measure the extent of recontamination during plucking resulting from feces escaping from the cloacae. The scalding kettle used had a volume of 90 L. The carcasses were dipped continuously and slowly into the scalding water over the immersion period. The scalding process ended when feathers on the chest could be plucked easily. Defeathering was performed using a rotating drum plucking machine for 10 s (drum diameter 930 mm, height 390 mm). Evisceration was done manually, in repetitive, uniform, predetermined movements, each time carried out by the same person. In total, processing each broiler took between 120 and 180 s from first to last sampling before stunning and after evisceration. Carcasses were transported manually between processing stations with clean gloves. After processing, the weight of internal organ bundles and whole carcasses were determined separately.

### 4.4. Scoring of Visible Contamination of Broiler Chickens

After arrival at the experimental slaughter facility, the breast skin area of broiler chickens was visually inspected concerning fecal contamination. Photographs were taken of the chickens in groups CG, 25kg/sqm, RR and CE and were then evaluated using a scoring system (Table 9). Photos for all groups were randomized and the scoring value was independently evaluated by six people having different scientific backgrounds (veterinary medicine, food technology, (molecular micro-)biology, physics). Mean values for both scoring categories (feathers, dirt), as well as the total sum for each broiler group, were calculated using IBM® SPSS® statistics (version 21). Data plots were examined using R (version 3.5.1).

### 4.5. Sampling

Contamination of the breast skin of the broiler chickens and the respective carcasses was investigated concerning ESBL/pAmpC-producing *Enterobacteriaceae* after arrival of the chickens at slaughter (ST1), after scalding (ST2), after defeathering (ST3) and after evisceration (ST4). Swab samples were taken from the breast skin of each of the 40 broiler chickens after each processing step. This was done by rubbing an area of 20 cm^2^ determined using a template (4 × 5 cm, COPAN Diagnostics, Murrieta, USA) with a cotton swab (Greiner Bio-One, Kremsmünster, Austria) moistened in Maximum Recovery Diluent (MRD; Merck, Darmstadt, Germany). Swab samples were stored on ice for transport to the laboratory. Processing of the samples was performed within 6 h of sampling.

### 4.6. Microbiological Analyses

Swab samples from all stations (ST1 to ST4) were cut into 3 mL of MRD each and vortexed extensively. Swab samples taken after arrival of the chickens (ST1) were additionally subjected to decimal dilutions prior to plating. Using an automated spiral plater, 100 µL of each sample (or each dilution) were plated onto MacConkey agar (Merck, Darmstadt, Germany) supplemented with 2 µg/mL Cefotaxime (MCCA-C) for quantification of CTX-resistant *Enterobacteriaceae*. Additionally, 100 µL of each sample and each dilution were plated on Violet Red Bile Dextrose Agar (VRBD; Merck, Darmstadt, Germany) for the determination of total *Enterobacteriaceae* counts. Plates were incubated aerobically at 37 °C for 18–24 h. CFU were determined using the spiral colony counting technique with a Whitley automatic spiral plater (Don Whitley Scientific, UK) and results were calculated as CFU/20 cm² breast skin. The limit of detection (LOD) for the quantification was calculated to be 30 CFU/20 cm² breast skin.

### 4.7. Statistical Analysis

Descriptive statistics were calculated using IBM^®^ SPSS^®^ statistics (version 21). Further analyses and plotting of data were performed using R (version 3.5.1) [31].

Analysis was conducted separately for CTX-resistant *Enterobacteriaceae* and overall number of *Enterobacteriaceae*. Bacteria counts and the common logarithm of the counts were inspected visually using histograms and Q-Q-plots for the different experimental groups. Furthermore, a Shapiro-Wilk test for normality was performed for overall and resistant *Enterobacteriaceae*. The visual inspection, Q-Q-plots and Shapiro-Wilk-test showed that most of the data could not be considered normally or lognormally distributed. Consequently, we chose nonparametric tests to further analyze this data. The Kruskal-Wallis test with a multiple contrast test as a following post hoc test were performed using the statistical software R version 3.5.1. The Kruskal-Wallis test was performed with the function kruskal.test() from the stats package and the multiple contrast test was performed employing the function mctp() from the nparcomp package [32]. Details on the statistical analyses are presented in the Appendix A. Linear models for dependencies between bacterial contamination, reduction during scalding and scalding temperature, were calculated using the software R. One model was based on the data for the CTX-resistant bacteria and one model for the data of total *Enterobacteriaceae*. In the Appendix A, the model for all *Enterobacteriaceae* is labeled model2 and the model for the resistant bacteria is labeled model3_res. Model selection was based on the analysis of the Bayesian Information Criterion (BIC).

## 5. Conclusions

In our study we investigated, in a small-scale experimental setting, the impact of on-farm interventions against CTX-resistant *Enterobacteriaceae* on the initial external contamination of broiler chickens, and its impact on the (cross) contamination of carcasses during slaughter. Furthermore, we evaluated a scoring system for the visible contamination of the broiler chickens with a mixture of feces, dirt and litter when entering the slaughter facility. We found that on-farm interventions can have an impact on the contamination level of the carcasses during processing, especially on the introduction of resistant bacteria into a slaughter facility. The investigated measures led to varying outcomes concerning the bacterial load on the breast skin as well as the general appearance of the broiler chickens (feathers/coat). In our study, a slow growing breed, reduced stocking density and competitive exclusion flora all showed significant effects on the bacterial contamination of the breast skin. Furthermore, broiler chickens at reduced stocking density had the best scores concerning visible contamination. Investigations concerning the interventions in our study were performed in small-scale experiments but showed significant differences between intervention groups and when compared to the control groups. The most promising measures found in our study provide evidence-based suggestions for possible interventions to be further evaluated on large-scale conventional broiler farms for their impact on the final contamination during subsequent processing in slaughterhouses. In addition, the evaluated scoring system could assist in identifying flocks at risk of increased fecal contamination on the skin or feathers, which might be a target for intervention.

## Figures and Tables

**Figure 1 antibiotics-10-00228-f001:**
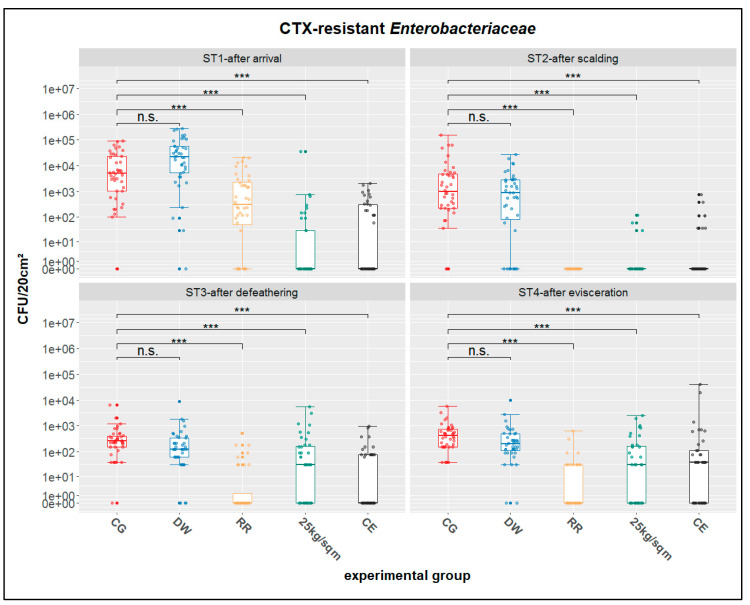
Distribution and statistics for CTX-resistant *Enterobacteriaceae*; ST1-after arrival, ST2-after scalding, ST3-after defeathering, ST4-after evisceration; CG-control group, DW-drinking water supplement, RR-slow growing breed Rowan × Ranger, 25kg/sqm-reduced stocking density, CE-competitive exclusion; results of nonparametric multiple contrast test are shown as: n.s.—not significant, ***—*p* < 0.001. The lower and upper hinges of the boxes correspond to the first and third quartiles (the 25th and 75th percentiles). The line in the box between the lower and upper hinges corresponds to the median. The upper whisker extends from the upper hinge to the largest value but no further than 1.5 × IQR from the hinge (where IQR is the inter-quartile range, or distance between the first and third quartiles). The lower whisker extends from the lower hinge to the smallest value but at most 1.5 × IQR of the hinge.

**Figure 2 antibiotics-10-00228-f002:**
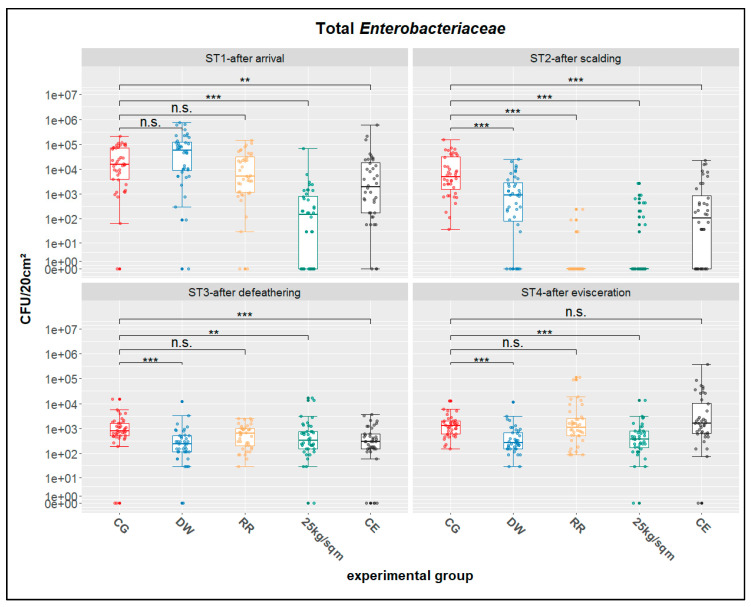
Distributions and statistics for total *Enterobacteriaceae*; ST1-after arrival, ST2-after scalding, ST3-after defeathering, ST4-after evisceration; CG-control group, DW-drinking water supplement, RR-slow growing breed Rowan × Ranger, 25 kg/sqm-reduced stocking density, CE-competitive exclusion; results of nonparametric multiple contrast test are shown as: n.s.—not significant, **—*p* < 0.01, ***—*p* < 0.001. The lower and upper hinges of the boxes correspond to the first and third quartiles (the 25th and 75th percentiles). The line in the box between lower and upper hinge corresponds to the median. The upper whisker extends from the upper hinge to the largest value but no further than 1.5 × IQR from the hinge (where IQR is the inter-quartile range, or distance between the first and third quartiles). The lower whisker extends from the lower hinge to the smallest value but at most 1.5 × IQR of the hinge.

**Figure 3 antibiotics-10-00228-f003:**
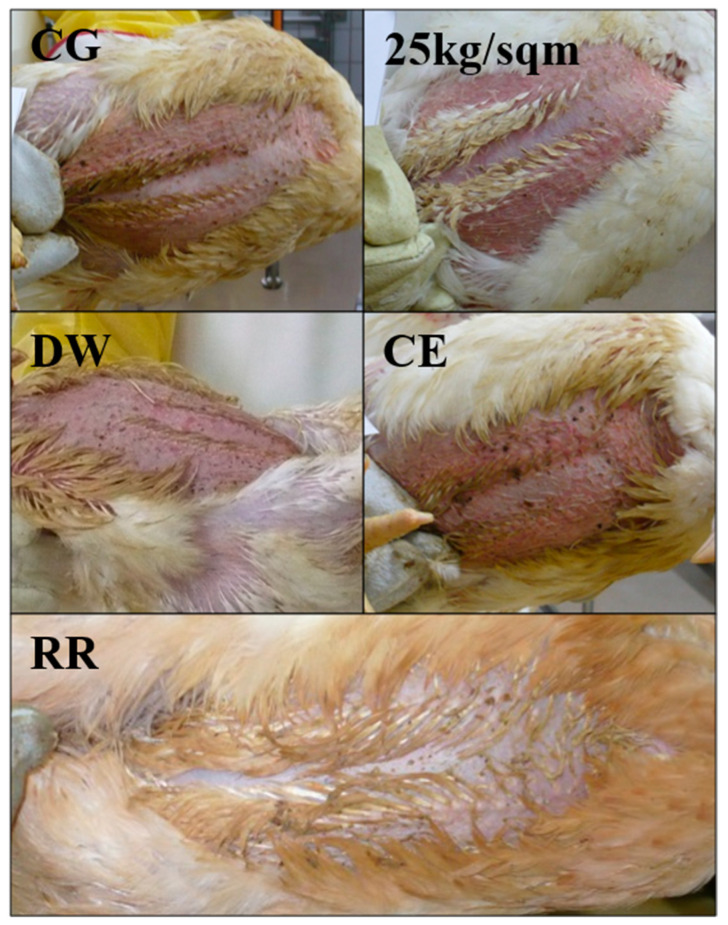
Visible fecal contamination inspection after arrival; CG-control group, DW-drinking water supplement, RR-slow growing breed Rowan × Ranger, 25 kg/sqm-reduced stocking density, CE-competitive exclusion.

**Figure 4 antibiotics-10-00228-f004:**
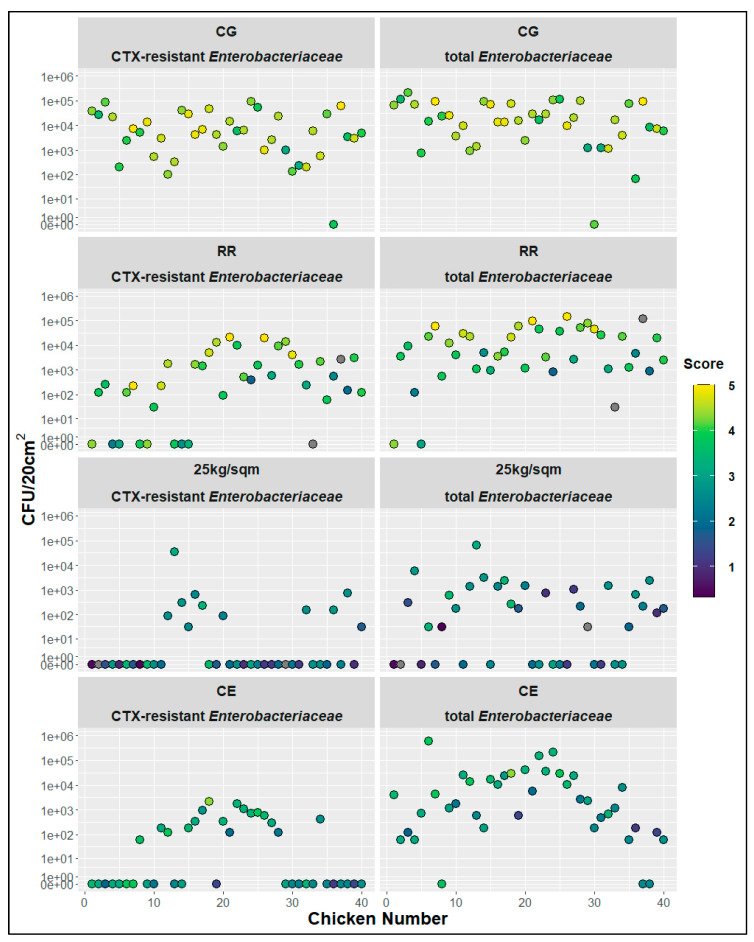
Chicken vise distribution of the total scoring values related to measures of bacterial counts for CTX-resistant and total *Enterobacteriaceae*, CG-control group, RR-slow growing breed Rowan × Ranger, 25 kg/sqm-reduced stocking density, CE-competitive exclusion; grey dots—not scored.

**Figure 5 antibiotics-10-00228-f005:**
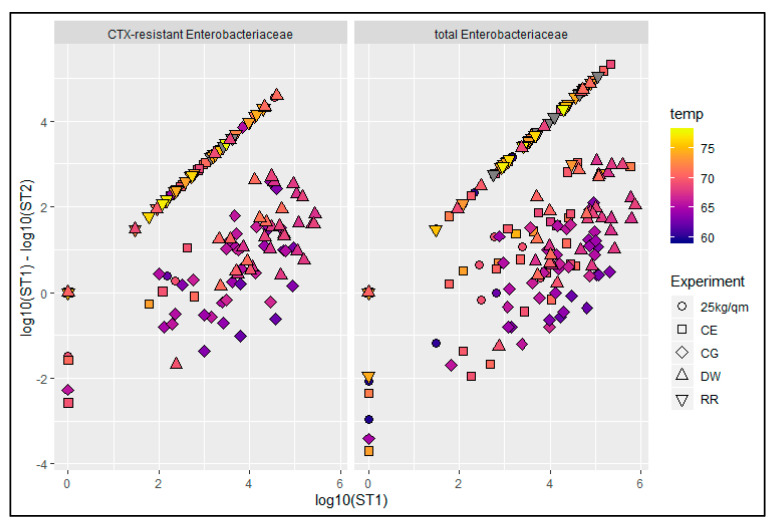
Plot of the log10 values of the initial concentration of CTX-resistant and total *Enterobacteriaceae* at station 1 (ST1) against the reduction of the bacterial counts after scalding (ST1-ST2). Scalding temperatures in °C (temp) are displayed in a color scale; CG-control group, DW-drinking water supplement, RR-slow growing breed Rowan × Ranger, 25 kg/sqm-reduced stocking density, CE-competitive exclusion; grey—temperature not determined.

**Figure 6 antibiotics-10-00228-f006:**
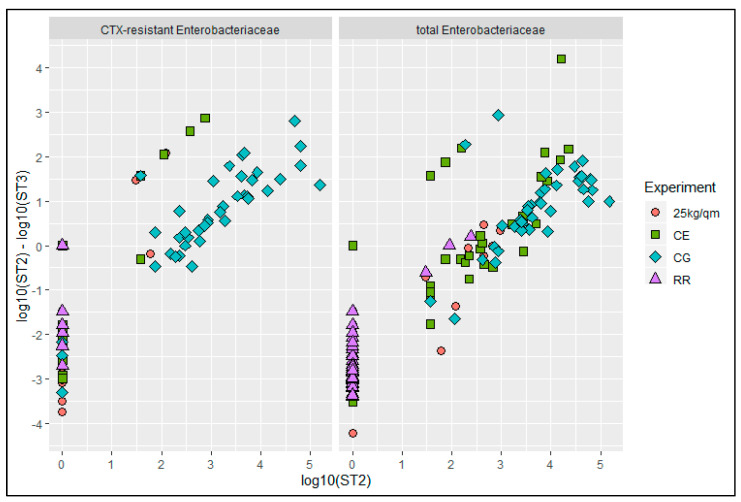
Plotting log10 values of numbers of CTX-resistant and total *Enterobacteriaceae* at station 2 (ST2, after scalding) against the reduction of the bacterial counts after defeathering (ST2-ST3). CG-control group, RR-slow growing breed Rowan × Ranger, 25 kg/sqm-reduced stocking density, CE-competitive exclusion.

**Table 1 antibiotics-10-00228-t001:** Animal parameters for all groups; CG-control group, DW-drinking water supplement, RR-slow growing breed Rowan × Ranger, 25 kg/sqm-reduced stocking density, CE-competitive exclusion.

Parameters	CG	DW	RR	25 kg/sqm	CE
**Breed**	Ross308	Ross308	Rowan × Ranger	Ross308	Ross308
**Fattening time**	34 days	38 days	47 days	34 days	34 days
**Mean weight carcasses**	2044 g	1934 g	1992 g	1889 g	1987 g
**Mean weight internal organ bundle**	239 g	238 g	239 g	257 g	266 g
**Outdoor temperatures during transportation**	Max 22 °C	Max 23 °C	Max 7 °C	Max 10 °C	Max 22 °C
**Month of slaughter**	April	August	November	March	October

**Table 2 antibiotics-10-00228-t002:** Qualitative and quantitative data (log10 CFU/20 cm² values) of the swab samples (*n* = 40 per sampling point) concerning Cefotaxime (CTX)-resistant *Enterobacteriaceae* for all groups and all samplings; CG-control group, DW-drinking water supplement, RR-slow growing breed Rowan × Ranger, 25 kg/sqm-reduced stocking density, CE-competitive exclusion.

	CG	DW	RR	25 kg/sqm	CE
**ST1, before slaughter**					
**No of positive samples**	39	39	31	11	17
**Mean log10 CFU/20 cm²**	4.21	4.72	3.46	2.98	2.41
**Median log10 CFU/20 cm²**	3.70	4.35	2.51	0	0
**ST2, after scalding**					
**No of positive samples**	37	32	0	3	7
**Mean log10 CFU/20 cm²**	4.03	3.47	0	0.72	1.56
**Median log10 CFU/20 cm²**	2.99	2.95	0	0	0
**ST3, after defeathering**					
**No of positive samples**	39	37	10	23	16
**Mean log10 CFU/20 cm²**	2.70	2.68	1.47	2.54	1.96
**Median log10 CFU/20 cm²**	2.42	2.08	0	1.48	0
**ST4, after evisceration**					
**No of positive samples**	40	38	13	21	23
**Mean log10 CFU/20 cm²**	2.84	2.77	1.54	2.38	3.21
**Median log10 CFU/20 cm²**	2.62	2.29	0	1.48	1.57

**Table 3 antibiotics-10-00228-t003:** Qualitative and quantitative data (log10 CFU/20 cm² values) of the swab samples (*n* = 40, each group and station) concerning total *Enterobacteriaceae* for all groups and all samplings; CG-control group, DW-drinking water supplement, RR-slow growing breed Rowan × Ranger, 25 kg/sqm-reduced stocking density, CE-competitive exclusion.

	CG	DW	RR	25 kg/sqm	CE
**ST1, before slaughter**					
**No of positive samples**	39	39	38	25	37
**Mean log10 CFU/20 cm²**	4.59	5.08	4.38	3.36	4.49
**Median log10 CFU/20 cm²**	4.20	4.77	3.72	2.18	3.30
**ST2, after scalding**					
**No of positive samples**	40	32	3	9	27
**Mean log10 CFU/20 cm²**	4.27	3.51	0.95	2.15	3.36
**Median log10 CFU/20 cm²**	3.70	2.96	0	0	2.05
**ST3, after defeathering**					
**No of positive samples**	38	39	40	39	35
**Mean log10 CFU/20 cm²**	3.21	2.85	2.88	3.12	2.80
**Median log10 CFU/20 cm²**	2.90	2.38	2.80	2.52	2.48
**ST4, after evisceration**					
**No of positive samples**	40	40	40	39	39
**Mean log10 CFU/20 cm²**	3.32	2.91	3.88	2.98	4.29
**Median log10 CFU/20 cm²**	3.11	2.43	3.05	2.57	3.21

**Table 4 antibiotics-10-00228-t004:** Means of scoring values for the broiler chickens in each experimental group averaged over all scorers. Mean minimum and maximum values were given in brackets. CG-control group, RR-slow growing breed Rowan × Ranger, 25 kg/sqm-reduced stocking density, CE-competitive exclusion.

Experimental Group.	Appearance Feathers/Coat	Dirt/Fecal Contamination	Total Sum Score
**CG**	2.51 (1.83–3.00)	1.81 (0.83–2.00)	4.32 (3.17–5.00)
**RR**	2.10 (0.83–3.00)	1.76 (0.17–2.00)	3.86 (1.83–5.00)
**25 kg/sqm**	1.15 (0.33–1.67)	1.03 (0.00–2.00)	2.18 (0.33–3.50)
**CE**	1.55 (0.50–2.50)	1.35 (0.33–2.00)	2.90 (1.17–4.33)

**Table 5 antibiotics-10-00228-t005:** Means of total sum scoring values for the broiler chickens for each of the six different scoring persons. CG-control group, RR-slow growing breed Rowan × Ranger, 25 kg/sqm-reduced stocking density, CE-competitive exclusion.

Scorer	1	2	3	4	5	6
**Experimental Group**						
**CG**	4.68	4.03	4.53	4.45	4.65	3.63
**RR**	4.21	3.58	4.00	4.00	4.29	3.05
**25 kg/sqm**	2.34	1.82	2.79	2.13	2.08	1.89
**CE**	3.48	2.50	3.50	2.85	2.68	2.40

**Table 6 antibiotics-10-00228-t006:** Mean bacterial concentrations (log10 CFU/20 cm²) after arrival (ST1) and after scalding (ST2), as well as the mean reduction via scalding for all groups. CG-control group, DW-drinking water supplement, RR-slow growing breed Rowan × Ranger, 25 kg/sqm-reduced stocking density, CE-competitive exclusion

Experimental Group	Mean Scalding Temperature (Range in °C)	CTX-Resistant *Enterobacteriaceae*			Total *Enterobacteriaceae*		
		Mean ST1	Mean ST2	Reduction (Ratio in %)	Mean ST1	Mean ST2	Reduction (Ratio in %)
CG	65 °C (71–78)	4.21	4.03	0.18 (33.93)	4.59	4.27	0.32 (52.19)
DW	69 °C (59–71)	4.72	3.47	1.25 (94.38)	5.08	3.51	1.57 (97.31)
RR	73 °C (68–74)	3.46	<LOD *	3.46 (>98.95)	4.38	0.95	3.43 (99.96)
25 kg/sqm	67 °C (63–67)	2.98	0.72	2.26 (99.45)	3.36	2.15	1.21 (93.83)
CE	71 °C (68–71)	2.41	1.56	0.85 (85.87)	4.49	3.36	1.13 (92.59)

* below limit of detection (LOD) of 30 CFU/20 cm².

**Table 7 antibiotics-10-00228-t007:** Results of two linear models for the log10-reduction of bacterial counts on carcass surfaces due to scalding. CG-control group, DW-drinking water supplement, RR-slow growing breed Rowan × Ranger, 25 kg/sqm-reduced stocking density, CE-competitive exclusion, ST1-after arrival, ST2-after scalding.

Dependent Variable: Log Reduction between ST1 and ST2						
	Model for Resistant *Enterobacteriaceae*			Model for all *Enterobacteriaceae*		
*Predictors*	*Estimates*	*CI*	*p*	*Estimates*	*CI*	*p*
**(Intercept)**	−21.34	−34.77–−7.91	**0.002**	−6.68	−11.28–−2.08	**0.005**
**temperature**	0.29	0.08–0.50	**0.006**	0.06	−0.01–0.13	0.105
**logST1**	0.84	0.7 –0.95	**<0.001**	0.81	0.68–0.94	**<0.001**
**DW**	−7.61	−26.57–1.36	0.430	1.09	0.51–1.66	**<0.001**
**RR**	22.63	3.56–41.69	**0.020**	2.93	2.11–3.76	**<0.001**
**25 kg/sqm**	21.55	7.4–35.62	**0.003**	2.64	2.05–3.23	**<0.001**
**CE**	26.44	8.90–43.98	**0.003**	1.39	0.76–2.02	**<0.001**
**temp:DW**	0.10	−0.18–0.39	0.478			
**temp:RR**	−0.30	−0.58–−0.03	**0.031**			
**temp:25kg/sqm**	−0.29	−0.51–−0.08	**0.008**			
**temp:CE**	−0.37	−0.63–−0.10	**0.006**			
Observations	190			190		
R^2^/R^2^ adjusted	0.684/0.666			0.621/0.608		

**Table 8 antibiotics-10-00228-t008:** Mean bacterial concentrations (log10 CFU/20 cm² values) after scalding (ST2) and after defeathering (ST3), as well as the mean reduction via defeathering for all groups. CG-control group, DW-drinking water supplement, RR-slow growing breed Rowan × Ranger, 25 kg/sqm-reduced stocking density, CE-competitive exclusion.

Experimental Group	CTX-Resistant *Enterobacteriaceae*			Total *Enterobacteriaceae*		
	Mean ST2	Mean ST3	Reduction	Mean ST2	Mean ST3	Reduction
CG	4.03	2.7	1.33	4.27	3.21	1.06
DW	3.47	2.68	0.79	3.51	2.85	0.66
RR	0	1.47	−1.47	0.95	2.88	−1.93
25 kg/sqm	0.72	2.54	−1.82	2.15	3.12	−0.97
CE	1.56	1.96	−0.4	3.36	2.8	0.56

**Table 9 antibiotics-10-00228-t009:** Developed scoring system for the evaluation of the level of visible external contamination as indicator for the microbiological contamination level.

Score	Appearance of Feathers/Coat	Score #	Dirt/Fecal Contamination
**0**	Chickens have a fluffy appearance with dry (white *) feathers.	**0**	There are no or at least very few small visible contaminations on the feathers or the breast skin.
**1**	Chickens still have some fluffy appearance with mostly dry (white *) feathers. Only some feathers are wet or slightly discolored.	**1**	There are few small contaminations of the feathers and the breast skin with only 1–2 large loosely attached contaminations on the coat.
**2**	There are wet or discolored feathers but mostly around breast skin area. Some of the feathers appear separated.	**2a**	Chickens are obviously contaminated on the breast skin and/or the coat (mainly small contaminations)
**3**	Large areas of the coat have wet or discolored feathers. Most of these feathers appear separated.	**2b**	Chickens are obviously contaminated on the breast skin and/or the coat (mainly large loosely attached contaminations)

* In case of Ross308; # categories 2a and 2b were calculated as 2 in the final scoring.

## Data Availability

The data presented in this study are available on request from the corresponding author.

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
