# Peer review of "Impact of On-Farm Interventions against CTX-Resistant Escherichia coli on the Contamination of Carcasses before and during an Experimental Slaughter"

_antibiotics, 2021, doi:10.3390/antibiotics10030228_

Round 1

Reviewer 1 Report

The manuscript by Projahn et al., “Impact of on-farm interventions against CTX-resistant Escherichia coli on the contamination of carcasses before and during an experimental slaughter,” presents a study that attempts to address the impact of several different rearing conditions on the contamination of chickens with ESBL/pAmpC producing E. coli and how this contamination propagates through the slaughtering process. This study presents interesting data that is undermined by the lack of appropriate experimental control, particularly with regard to the conditions in-which the birds were raised (individual experimental groups were raised at discrete times during the year, resulting in highly variable ambient conditions, especially temperature and day/night cycle) and the temperature of the scalding step which was highly variable between test groups. The failure to control these conditions makes interpreting differences between the on-farm interventions difficult.

Major concerns:

Introduction:

What is the rational for the on-farm interventions having an impact on contamination and recontamination during processing? Is this rational independent of the initial colonization level at ST1? If colonization at ST1 is the prime driver of colonization at ST2-4 then after ST1, the birds should be sorted at ST1 by colonization level to address this question.

Results:

What hypotheses are each experimental group meant to test and why were these specific conditions selected?

The fact that the various groups were grown at different times of the year is a critical problem of this study as in addition to external temperature, which clearly varied dramatically according to Table 2, the humidity and length of daylight would also be highly variable from November to August. These factors could have profound impacts on bacterial growth and spread and bird behavior and activity patterns. The potential impact of these factors on the results and conclusions of this study needs to be more directly and explicitly addressed. To conduct this experiment properly, groups of birds from each experimental intervention should have been tested at each time point.

The use of atypically high scalding temperatures for this study was a very poor choice in the experimental design of this study. If studying recontamination were the primary goal of this study then it might be a reasonable choice, but as the goal of the study was to assess the impact of on farm interventions on the number ESBL/pAmpC expressing bacteria present at each step of the slaughtering process, then production relevant scalding temperatures should have been used. A separate experiment specifically studying recontamination after scalding should have been designed and carried out if that was a desirable factor to assess.

Overall, too many variables were changed in each of the experimental groups to effectively attribute causality to any of the specific interventions.

Discussion:

The impact of rearing the birds at different times of year and the subsequent impact on growth conditions needs to be addressed. These differing rearing conditions could impact bird behavior and bacterial growth and colonization, which could in turn be responsible for the differences observed between the interventions.

Minor concerns:

Introduction:

Define competitive exclusion and why it might work.

What kind of drinking water additives are used?

It is generally better practice no to claim a complete absence of work addressing a particular topic. Say “little is known” rather than “currently no studies are published”

“Furthermore, also the magnitude” should be “Furthermore, the magnitude”

“Furthermore, we evaluated the steps ….” This sentence is confusingly worded, particularly given that it appears from some of the data that some processing steps increase the degree of contamination.

The rational behind implementing the visual scoring system should be presented.

What previous data exists that would suggest that the tested interventions will reduce ESBL/pAmpC producing bacterial contamination.

Results:

Better define the experimental groups here in addition to in the methods.

More information about the CE strain used in this study would be valuable.

Was there any effort made to measure colonization of the CE group with the probiotic strain during processing?

Keep the order of the various treatments the same throughout the tables. Comparing data between tables when the groups are in different orders in the tables is frustrating.

The variation in mean scalding temperature needs to be displayed in tables 2 and 3.

The use of the Conover-Iman test on the data in figure 1 is unnecessary and that test is not sufficiently conservative to provide useful conclusions from this data set.

The value of the feather condition and fecal contamination scoring system to the current paper is not clear.

What does: “scalding temperature were measured via a data logger or thermometer” mean?

Reporting the CFU reductions after scalding as ratios would be very informative and would allow easier comparisons within and between groups.

Discussion:

Overall, the English in the paper is reasonably acceptable, but the language use in the discussion is not acceptable and needs to be edited to improve grammar and clarity.

Once the birds have arrived at the processing facility, if the subsequent data were controlled for initial colonization levels at ST1, is there any reason to believe that the on-farm interventions would have different effects on subsequent contamination levels?

Author Response

The manuscript by Projahn et al., “Impact of on-farm interventions against CTX-resistant Escherichia coli on the contamination of carcasses before and during an experimental slaughter,” presents a study that attempts to address the impact of several different rearing conditions on the contamination of chickens with ESBL/pAmpC producing E. coli and how this contamination propagates through the slaughtering process. This study presents interesting data that is undermined by the lack of appropriate experimental control, particularly with regard to the conditions in-which the birds were raised (individual experimental groups were raised at discrete times during the year, resulting in highly variable ambient conditions, especially temperature and day/night cycle) and the temperature of the scalding step which was highly variable between test groups. The failure to control these conditions makes interpreting differences between the on-farm interventions difficult.

  • We thank the reviewer for the intensive reading and the comments made on the manuscript. However, we have to note that chickens were raised in an experimental animal facility “under identical conditions independently from the year´s season” as stated in line 442.
  • We agree that the scalding process in our experimental facility should be improved. Therefore, results from the reduction of the microbiological load during scalding were not set into relation with farm interventions but the individual contamination level of each carcass. This was described in lines 240 - 249

Major concerns:

Introduction:

What is the rational for the on-farm interventions having an impact on contamination and recontamination during processing? Is this rational independent of the initial colonization level at ST1? If colonization at ST1 is the prime driver of colonization at ST2-4 then after ST1, the birds should be sorted at ST1 by colonization level to address this question.

  • The rationale of the study was stated in lines 65-71. Reducing the contamination with resistant bacteria on farm-level results in a reduced entry of these resistant bacteria into the slaughter process and consequently a reduced contamination of the raw meat sold to the consumer. On-farm measurements might have a direct impact on the external and internal colonization. The external contamination was detected on ST1, the internal colonization has an impact on the re-contamination during processing due to the release of fecal material. It´s not possible to sort the chickens according to the colonization level, as the levels can change within days before the results are available in the laboratory.

Results:

What hypotheses are each experimental group meant to test and why were these specific conditions selected?

  • Tested farm factors aimed to reduce the colonization of ctx-resistant bacteria. The investigation of a respective outcome on farm-level was not part of this study. This can be found at Robé et al.

The fact that the various groups were grown at different times of the year is a critical problem of this study as in addition to external temperature, which clearly varied dramatically according to Table 2, the humidity and length of daylight would also be highly variable from November to August. These factors could have profound impacts on bacterial growth and spread and bird behavior and activity patterns. The potential impact of these factors on the results and conclusions of this study needs to be more directly and explicitly addressed. To conduct this experiment properly, groups of birds from each experimental intervention should have been tested at each time point.

  • We agree with the reviewer but in this study chickens were raised under identical conditions as stated in line 442. Table 1 was adjusted.

The use of atypically high scalding temperatures for this study was a very poor choice in the experimental design of this study. If studying recontamination were the primary goal of this study then it might be a reasonable choice, but as the goal of the study was to assess the impact of on farm interventions on the number ESBL/pAmpC expressing bacteria present at each step of the slaughtering process, then production relevant scalding temperatures should have been used. A separate experiment specifically studying recontamination after scalding should have been designed and carried out if that was a desirable factor to assess.

  • The study was conduced to investigated the impact of the farm-interventions on the external contamination on ST1. Furthermore, the impact on the further processing was investigated with special focus on recontamination. Therefore, scalding temperatures were set at a higher level than typical. A mayor factor on recontamination is the release of fecal material during processing. This in reverse conclusion is dependent on the colonization level and the possible impact of a farm-intervention.

Overall, too many variables were changed in each of the experimental groups to effectively attribute causality to any of the specific interventions.

  • As mentioned above chickens were raised under identical conditions.

Discussion:

The impact of rearing the birds at different times of year and the subsequent impact on growth conditions needs to be addressed. These differing rearing conditions could impact bird behavior and bacterial growth and colonization, which could in turn be responsible for the differences observed between the interventions.

  • This was already addressed above. Chickens were housed under identical conditions independently from the year´s season.

Minor concerns:

Introduction:

Define competitive exclusion and why it might work.

  • The respective application of the farm-inverventions were not part of the study. A detailed description of each of the applied farm interventions would extent the introduction atypically, but can be found here (Robe at al.). Respective references to CE were given.

What kind of drinking water additives are used?

  • The drinking water supplement is a mix of organic acids widely used in broiler production as stated in lines 451-453.

It is generally better practice no to claim a complete absence of work addressing a particular topic. Say “little is known” rather than “currently no studies are published”

  • We do not claim the complete absence of the data. It might be possible that respective data are available; however, they were not published in a scientific manner that one could get access to it as mentioned in the referenced review.

“Furthermore, also the magnitude” should be “Furthermore, the magnitude”

  • done

“Furthermore, we evaluated the steps ….” This sentence is confusingly worded, particularly given that it appears from some of the data that some processing steps increase the degree of contamination.

  • The sentence was rephrased (lines 69/70)

The rational behind implementing the visual scoring system should be presented.

  • Further information were given in lines 163-167

What previous data exists that would suggest that the tested interventions will reduce ESBL/pAmpC producing bacterial contamination.

  • The detailed evaluation of the respective interventions were part of a separate study (Robe et al.). Respective references were given, some interventions seem to be not yet tested.

Results:

Better define the experimental groups here in addition to in the methods.

  • Additional information were added in lines 78-80

More information about the CE strain used in this study would be valuable.

  • A patent application is running on that strain currently. Therefore it´s not possible to give further information on that strain.

Was there any effort made to measure colonization of the CE group with the probiotic strain during processing?

  • This is an interesting point but it was not possible to do respective examinations in this part of the project.

Keep the order of the various treatments the same throughout the tables. Comparing data between tables when the groups are in different orders in the tables is frustrating.

  • Tables were adjusted

The variation in mean scalding temperature needs to be displayed in tables 2 and 3.

  • Table 2 and 3 display numbers of bacteria determined at each stage of the slaughter process. Scalding temperatures are stated in table 6 were reductions during scalding are displayed. Temperature ranges were added in table 6.

The use of the Conover-Iman test on the data in figure 1 is unnecessary and that test is not sufficiently conservative to provide useful conclusions from this data set.

  • As described in the caption of Figure 1 it is not the results of the Conover-Iman test, which are displayed in the plot, but the results of the multiple contrast test. Even if we had used the Conover-Iman test with Bonferroni correction the result would have been the same. As discussed in the statistical supplement four of the five post-poc tests performed (Dunn test with Bonferroni and Holm correction, Conover-Iman test with Bonferroni and Holm correction and the multiple contrast test) gave the same result: The water group does not significantly differ from the control group, all other groups do differ significantly. Only the Conover-Iman test with Holm correction - the least conservative test in out battery of tests - shows a significant difference between control and water group throughout all stations. But this “extreme” result is not shown in Figure 1.

The value of the feather condition and fecal contamination scoring system to the current paper is not clear.

  • If a bad feather condition is related to the fecal contamination, it is an easy method applicable for farmers or slaughterhouses comparable with scoring systems for feet pad lesions etc. on the chicken health.

What does: “scalding temperature were measured via a data logger or thermometer” mean?

  • Temperatures were measured with a temperature data logger or a thermometer.

Reporting the CFU reductions after scalding as ratios would be very informative and would allow easier comparisons within and between groups.

  • Ratios were added in table 6

Discussion:

Overall, the English in the paper is reasonably acceptable, but the language use in the discussion is not acceptable and needs to be edited to improve grammar and clarity.

  • The manuscript was revised concerning language style.

Once the birds have arrived at the processing facility, if the subsequent data were controlled for initial colonization levels at ST1, is there any reason to believe that the on-farm interventions would have different effects on subsequent contamination levels?

  • Yes, as a major factor in (re-)contamination during slaughter is the release of fecal material. Consequently, the internal colonization affected by the intervention measurement has an impact on the slaughter process.

Reviewer 2 Report

Review of Impact of on-farm interventions against CTXresistant Escherichia coli on the contamination of carcasses before and during an experimental slaughter for ANTIBIOTICS

This study looks for ways to decrease the contamination of carcasses by antibiotic resistant bacteria. Overall, the data is interesting but there is a need to make the MS shorter and illustrate clearly the impacts that are observed.

  • The abstract needs to contain quantitative numbers for a quick assessment of each factor
  • The paper should be edited for clarity (tables should have superscript and a column for statistics, figures should plot the factor of interest as the x axis instead of relying on the colors)
  • Why do you use 2 strains ? Can you monitor each CTX-resistant strain at slaughter ?
  • For the CE group, more information is needed on the strain IHIT36098, and it should be clearly stated in the abstract and in the results (the reader learns about the exact strain on page 15). Why do you use a single strain when complex consortia exists and are more efficient (Undefined and Defined Bacterial Preparations for the Competitive Exclusion of Salmonella in Poultry - A Review)?

Other details :

Table 1: Animal parameters instead of (animal) parameter

The competitive exclusion product should be clearly mentioned in the abstract and the results sections.

Table 2 should contain an additional column with p-values and superscripts to see the differences that are significant after the posthoc tests. It could be given as suppl data, since Fig 1 shows the same information in a more meaningful manner.

Table 3 should contain an additional column with p-values and superscripts to see the differences that are significant after the posthoc tests. It could be given as suppl data, since Fig 2 shows the same information in a more meaningful manner. Also please mention that the tables report the logs of the values.

All the tables could benefit from additional statistical information and be given as suppl data to avoid redundancy of with the figures.

Why are you using non parametric tests here ?

On fig1, please add ST1-after arrival instead of ST1 on the figure for an easier read.

Check italics for Enterobacteriaceae

I don’t understand the point of showing the chicken number in a fig 4.

Fig 5 isnot a straigt forward presentation of your point. Can you imagine a graph of the decrease vs the temp to make your point clearer ?

Table 7 is good, but should be combined with the measurements in a single table.

Author Response

We would like to thank the reviewers for the intensive reading and the comments made on the manuscript. Please find below our answers.

Review of Impact of on-farm interventions against CTXresistant Escherichia coli on the contamination of carcasses before and during an experimental slaughter for ANTIBIOTICS

This study looks for ways to decrease the contamination of carcasses by antibiotic resistant bacteria. Overall, the data is interesting but there is a need to make the MS shorter and illustrate clearly the impacts that are observed.

  • The abstract needs to contain quantitative numbers for a quick assessment of each factor

Unfortunately, the abstract is limited to 200 words at maximum. Therefore, we had to shorten all the outcomes and data. However, we made some modifications for improved clarity.

  • The paper should be edited for clarity (tables should have superscript and a column for statistics, figures should plot the factor of interest as the x axis instead of relying on the colors)

Because of the high amount of data and statistical analyses we added a respective appendix where all the statistical calculations were explained.

  • Why do you use 2 strains ? Can you monitor each CTX-resistant strain at slaughter ?

Two strains were used because of the different type of CTX-resistance. AmpC producing E. coli is more likely found in broiler chickens. However, ESBL-producing E. coli is more often investigated. It is not possible to differentiate both strains on the agar plated we used. However, we wanted to determine all resistant bacteria including strains that might got resistant by plasmid transfer during the fattening period. The initial decision for the two strains was made by the researchers responsible for the first part of the project (Robe et al)

  • For the CE group, more information is needed on the strain IHIT36098, and it should be clearly stated in the abstract and in the results (the reader learns about the exact strain on page 15). Why do you use a single strain when complex consortia exists and are more efficient (Undefined and Defined Bacterial Preparations for the Competitive Exclusion of Salmonella in Poultry - A Review)?

We added the information on the strain in the abstract and results section on page 2. A single strain was used as it turned out that in contrast to Salmonella for ESBL/AmpC-producing E. coli the application of an undefined CE preparation results in various outcomes depending on the study and the strains used as discussed in lines 397-402

Other details :

Table 1: Animal parameters instead of (animal) parameter

done

The competitive exclusion product should be clearly mentioned in the abstract and the results sections.

We added the information on the strain in the abstract and results section on page 2.

Table 2 should contain an additional column with p-values and superscripts to see the differences that are significant after the posthoc tests. It could be given as suppl data, since Fig 1 shows the same information in a more meaningful manner.

It is not clear which p-values should be included in Table 2. We did not perform tests on the differences of the means, therefore we do not have for each group`s mean/median a p-value. The Kruskal-Wallis test we performed might be considered a test whether the medians of the groups are the same but only if one assumes that the distribution in each group is the same. This assumption seems not to be met when looking at the distributions depicted on page 4 and page 8 of the statistical supplement. Therefore, it seems that the result of the Kruskal-Wallis test in our case just provides evidence that the mean ranks of the groups are not all the same. To report the mean ranks of the groups in the table and then putting at each rank the same p-value (namely the p-value obtained from the Kruskal-Wallis test) would be an option but certainly not a common one. Table 2 is just a summary of purely descriptive statistics that give an impression of the form of the data without going into statistical inferences about differences.

Table 3 should contain an additional column with p-values and superscripts to see the differences that are significant after the posthoc tests. It could be given as suppl data, since Fig 2 shows the same information in a more meaningful manner. Also please mention that the tables report the logs of the values.

As stated in our previous comment we are critical of providing p-values in Table 2 and Table 3. Now that the reviewer stated that he is referring to the p-values from the post-hoc test we see the problem of confusion. Tables 2 and Table 3 show means and medians. Providing the p-values from the post-hoc test in these Tables might lead the reader to think that the p-value belongs to tests testing for differences in means or medians. But, as mentioned in the previous comment that is not the case. The p-values of the post-hoc test, codified in stars as shown in Figure 1 and Figure 2, are from nonparametric pairwise tests between the control group and each individual experimental group for stochastic dominance of one or the other group. The p-values are not derived from direct test for difference of means of medians. As recested, log values were added in table 2 and 3.

All the tables could benefit from additional statistical information and be given as suppl data to avoid redundancy of with the figures.

Data on statistical test can be found in the appendix

Why are you using non parametric tests here ?

As described in sections 1.1.1 and 1.1.2 of the statistical supplement we used nonparametric tests here because we came to the conclusion that the data could not be considered normally distributed. We came to this conclusion after inspecting the data visually in terms of distributions and Q-Q-plots and after performing Shapiro-Wilk tests for normality. Therefore we opted for a more cautious and conservative approach to data analysis, i.e. nonparametric tests.

On fig1, please add ST1-after arrival instead of ST1 on the figure for an easier read.

Requested changes were made in figure 1. Figure 2 was adjusted respectively.

Check italics for Enterobacteriaceae

done

I don’t understand the point of showing the chicken number in a fig 4.

What would you recommend instead?

Fig 5 isnot a straigt forward presentation of your point. Can you imagine a graph of the decrease vs the temp to make your point clearer ?

As we wanted to show in the first place that, the reduction during scalding seemed to be dependent on the initial concentration, in our opinion ST1 should be placed on the x-axis lines 442-443. A possible dependency on the temperature was integrated via the coloring.

Table 7 is good, but should be combined with the measurements in a single table.

Table 6 and 7 can not be combined as table 6 represents mean values only. Table 7 shows results from the models based on 190 separate observations. As stated in lines 244-253 we did not find correlations based on the mean values. Therefore, adding these values to table 7 would lead to confusing results/outcomes.

Round 2

Reviewer 1 Report

The authors have done an acceptable job accepting the relevant revisions.

Author Response

We thank the reviewer for the intensive reviewing of the manuscript.